# Synaptoproteomic Analysis of the Prefrontal Cortex Reveals Spatio-Temporal Changes in SYNGAP1 Following Cannabinoid Exposure in Rat Adolescence

**DOI:** 10.3390/ijms24010698

**Published:** 2022-12-31

**Authors:** Johanna S. Qvist, Maria Scherma, Nitya Jayaram-Lindström, Walter Fratta, Denise B. Kandel, Eric R. Kandel, Paola Fadda, Philippe A. Melas

**Affiliations:** 1Department of Neuroscience, Columbia University, New York, NY 10032, USA; 2Mortimer B. Zuckerman Mind Brain Behavior Institute, Jerome L. Greene Science Center, New York, NY 10027, USA; 3Department of Biomedical Sciences, Division of Neuroscience and Clinical Pharmacology, University of Cagliari, 09042 Cagliari, Italy; 4Center for Psychiatry Research, Department of Clinical Neuroscience, Karolinska Institutet & Stockholm Health Care Services, 11364 Stockholm, Sweden; 5Department of Psychiatry, Vagelos College of Physicians and Surgeons and Mailman School of Public Health, Columbia University, New York, NY 10032, USA; 6CNR Institute of Neuroscience, 09042 Cagliari, Italy; 7Center for Molecular Medicine, L8:00, Karolinska University Hospital, 17176 Stockholm, Sweden

**Keywords:** cannabis, THC, PFC, synaptic Ras-GAP 1, SYNGAP, MRD5, RASA1

## Abstract

The regular use of cannabis during adolescence has been associated with a number of negative life outcomes, including psychopathology and cognitive impairments. However, the exact molecular mechanisms that underlie these outcomes are just beginning to be understood. Moreover, very little is known about the spatio-temporal molecular changes that occur following cannabinoid exposure in adolescence. To understand these changes, we exposed mid-adolescent male rats to a synthetic cannabinoid (WIN 55,212-2 mesylate; WIN) and, following drug abstinence through late adolescence, we subjected the synaptosomal fractions of the prefrontal cortex (PFC) to proteomic analyses. A total of N = 487 differentially expressed proteins were found in WIN-exposed animals compared to controls. Gene ontology analyses revealed enrichment of terms related to the gamma-aminobutyric acid (GABA)-ergic neurotransmitter system. Among the top differentially expressed proteins was the synaptic Ras GTPase-activating protein 1 (SYNGAP1). Using Western blotting experiments, we found that the WIN-induced upregulation of SYNGAP1 was spatio-temporal in nature, arising only in the synaptosomal fractions (not in the cytosol) and only following prolonged drug abstinence (not on abstinence day 1). Moreover, the SYNGAP1 changes were found to be specific to WIN-exposure in adolescence and not adulthood. Adolescent animals exposed to a natural cannabinoid (Δ^9^-tetrahydrocannabinol; THC) were also found to have increased levels of SYNGAP1 in the PFC. THC exposure also led to a pronounced upregulation of SYNGAP1 in the amygdala, but without any changes in the dorsal striatum, hippocampus, or nucleus accumbens. To our knowledge, this is the first study to uncover a link between cannabinoid exposure and changes in SYNGAP1 that are spatio-temporal and developmental in nature. Future studies are needed to investigate the putative role of SYNGAP1 in the negative behavioral consequences of cannabis use in adolescence.

## 1. Introduction

Regular cannabis use among adolescents has been associated with a number of negative life outcomes, including depression, substance use disorders, and psychosis, as well as lower educational attainment, lower income, and lower levels of life satisfaction [1,2,3,4]. Moreover, early use of cannabis has been associated with altered neurodevelopment, particularly in prefrontal cortical (PFC) regions undergoing age-related thickness changes [5]. These findings have suggested that psychoactive cannabis components, such as Δ^9^-tetrahydrocannabinol (THC), exert pathophysiological effects by interfering with the brain’s endogenous cannabinoid, i.e., its endocannabinoid, system. The endocannabinoid system consists of lipid-based retrograde neurotransmitters, such as anandamide (AEA) and 2-arachidonoylglycerol (2-AG), that bind to cannabinoid receptors and its disturbance by cannabis exposure or stressful life events has been linked to pathophysiology and aberrations in neurodevelopmental processes [6,7,8,9,10,11,12].

Despite the consistent associations between adolescent cannabis use and increased likelihood of negative life outcomes, little is known about the exact molecular pathways that underlie the observed behavioral phenotypes. Most molecular evidence, to date, originates from animal models where cannabinoids are administered to rodents in adolescence. These studies have reported neurochemical changes that dysregulate developmental processes and have implicated cortical neurotransmitter systems utilizing gamma-aminobutyric acid (GABA) and glutamate [13,14,15,16]. For instance, a functional downregulation of GABAergic neurotransmission was reported in the adult rodent PFC following adolescent exposure to both natural and synthetic psychoactive cannabinoids, such as THC and WIN 55,212-2 mesylate (WIN), respectively [17,18]. Exposure to WIN during rat adolescence has also been found to cause a synaptic decrease of glutamate receptors in the PFC following drug abstinence [14]. Moreover, human adolescent cannabis users have been found to have reduced GABA and glutamate levels in cortical regions [19,20].

In the present study, we sought to provide additional insights into the molecular changes that occur as a result of cannabinoid exposure in adolescence, utilizing an unbiased approach, with a focus on synaptic fractions and the PFC. To this end, we treated male rats with cannabinoids in adolescence, and then subjected the PFC to synaptoproteomics and examined the presence of spatio-temporal differences in identified proteins. Our data uncovered the first evidence, to our knowledge, of a causal relationship between cannabinoid exposure and aberrations in levels of the synaptic Ras GTPase-activating protein 1 (SYNGAP1) that is spatio-temporal and developmental in nature. Specifically, changes in SYNGAP1 were found to be present only in the synaptic fractions (not in the cytosol), only following prolonged drug abstinence (not on abstinence day 1), and only when cannabinoids were administered in adolescence (not in adulthood). Since aberrations in the *SYNGAP1* gene have been linked to neurodevelopmental disorders [21], we discuss the possibility of SYNGAP1 being a key mediator of the association between adolescent cannabis use and adverse neuropsychiatric outcomes [22].

## 2. Results

### 2.1. Synaptoproteomics Reveal GABAergic Changes in the PFC following WIN Exposure in Adolescence

Since puberty is a vulnerable developmental period for the consequences of cannabis exposure [23], we utilized an unbiased synaptoproteomic approach to identify synaptic protein changes that occur in the male rat PFC in late adolescence (i.e., PND 64; [24]) following exposure to a synthetic cannabinoid (WIN) in mid-adolescence/puberty (i.e., PND 42–52; [24]). A total of N = 487 differentially expressed proteins were found in WIN-exposed animals compared to controls (ANOVA, q < 0.01; Figure 1 and Appendix A). In line with previous reports demonstrating an effect of cannabinoids on GABAergic neurotransmission [17,18], gene ontology (GO) analyses of the differentially expressed proteins in our study showed enrichment for biological processes involving GABA, including GABA biosynthetic process (GO:0009449), GABA metabolic process (GO:0009448), GABAergic synaptic transmission (GO:0051932), and GABA signaling pathway (GO:0007214) (q < 0.05; Appendix A). Other highly enriched terms in the GO analyses included the regulation of long-chain fatty acid import into cell (GO:0140214), amygdala development (GO:0021764) and regulation of the neurotrophin TRK receptor signaling pathway (GO:0051388) (q < 0.05; Appendix A).

### 2.2. Exposure to WIN in Adolescence Results in Spatio-Temporal Changes in SYNGAP1

Next, to identify individual molecules that may be responsible for the aberrations in GABAergic signaling revealed by the GO analyses, we examined the list of proteins with significant differences between WIN-exposed animals and controls (Appendix A). In line with previous literature [16,17,18], there were changes in GABA receptors and their subunits, including GABRB1, GABRA5, GABRD, GABRA3 and GABBR1 (Appendix A). However, we also observed a novel and highly significant change in levels of the synaptic Ras GTPase-activating protein 1 (SYNGAP1, q = 0.00000679; Appendix A), another molecule with a key role in GABAergic neurotransmission [25,26,27]. To verify the changes in SYNGAP1 revealed by the proteomic analyses, we performed Western blotting experiments using synaptosomal PFC extracts. These experiments confirmed a significant upregulation of SYNGAP1 on drug abstinence day (AD) 12 following WIN exposure in mid-adolescence (*t*-test, *p* = 0.0028; Figure 2A). Since SYNGAP1 is known to also localize in non-synaptic compartments, e.g., in the cytosol [28,29], we examined putative changes in the cytosolic fractions of the PFC. However, no changes were found in the cytosol (*t*-test, *p* = 0.998; Figure 2B), suggesting that the observed changes are spatial in nature and specific to the synaptic compartments. Next, we asked whether the changes in SYNGAP1 may be time-dependent relative to the last drug exposure. To this end, we performed Western blotting experiment using synaptosomal PFC fractions from adolescent rats that were sacrificed one day after the last WIN administration, i.e., on AD 1. Results from this experiment showed no differences in levels of SYNGAP1 (*t*-test, *p* = 0.498; Figure 2C), suggesting that the observed changes are also temporal in nature and arise following longer drug abstinence.

### 2.3. No Changes in Synaptic Levels of SYNGAP1 When WIN Is Administered in Adulthood

Next, we asked whether the observed WIN-associated synaptic changes in levels of SYNGAP1 are specific to adolescence. To this end, we treated adult animals with WIN and sacrificed them either 12 days or one day following the last WIN exposure, i.e., on AD 12 or AD 1, to mimic the time points assessed in adolescence. Western blotting experiments using synaptosomal PFC extracts from adult animals revealed no changes in levels of SYNGAP1 at either AD 12 (*t*-test, *p* = 0.383; Figure 3A) or AD 1 (*t*-test, *p* = 0.29; Figure 3B), suggesting that the effects of WIN-exposure on levels of SYNGAP1 are specific to adolescence.

### 2.4. Exposure to THC in Adolescence affects SYNGAP1 Levels at Specific Brain Regions

Finally, we asked whether the observed synaptic changes in levels of SYNGAP1 are (i) specific to WIN or if they also occur following exposure to THC, and (ii) specific to the PFC or if they also occur in other brain regions. To this end, adolescent animals were treated with THC, and synaptosomal fractions from five brain regions were isolated on abstinence day 12. Western blotting experiments revealed significant upregulation of SYNGAP1 in the PFC of THC-treated animals (two-way ANOVA, Treatment x Brain Region interaction: F (4, 52) = 114.9, *p* < 0.0001; Sidak’s multiple comparison test for PFC, *p* = 0.0198; Figure 4). A highly pronounced upregulation of SYNGAP1 was also found in the amygdala of THC-treated animals (Sidak’s multiple comparison test for AMYG, *p* < 0.0001; Figure 4). No significant changes in SYNGAP1 levels were found in the dorsal striatum, hippocampus, or nucleus accumbens of THC-treated animals (Sidak’s multiple comparison tests, *p* > 0.05; Figure 4). These data suggest that both natural and synthetic psychoactive cannabinoids affect synaptic levels of SYNGAP1 when administered in adolescence, and that these effects are brain region-specific.

## 3. Discussion

Psychoactive cannabis constituents interfere with normal endocannabinoid signaling and affect neural pathways that regulate anxiety, reward, and cognition [7,30,31]. Adolescence, and in particular puberty, has been defined as a highly vulnerable developmental period for the deleterious consequences of cannabis use on cognitive function and mental health [1,23,32]. Neuroimaging studies of adolescent cannabis users have implicated frontoparietal regions, including the prefrontal cortex (PFC), in brain activation changes related to reward, memory, and inhibitory control [33]. However, the molecular mechanisms that underlie these changes remain elusive. Thus, in the present preclinical study, we exposed rats to a synthetic cannabinoid (WIN) during mid-adolescence/puberty and isolated synaptosomes from the PFC following a 12-day period of forced drug abstinence. Synaptosomal extracts were then subjected to unbiased proteomic analyses that revealed a total of N = 487 differentially expressed proteins, between WIN-treated animals and controls, and with GO analyses showing an enrichment for biological processes involving GABA. This is in line with previous studies reporting aberrations in GABAergic neurotransmission in the adult rat PFC following adolescent exposure to psychoactive cannabinoids [17,18], as well as GABA abnormalities in cortical regions of human adolescent cannabis users [19]. Critically, aberrations in the GABAergic system that occur in adolescence, e.g., through the use of exogenous cannabinoids, can lead to cognitive abnormalities that persist into adulthood [15].

We also found that one of the most significant differentially expressed proteins was the synaptic Ras GTPase-activating protein 1 (SYNGAP1), an excitatory synapse-specific Ras GTPase activating protein that regulates silent synapse formation [34] and modulates GABAergic synaptic function [25,27]. Importantly, pathogenic mutations in the *SYNGAP1* gene have been found to disrupt the maturation of dendritic spine synapses and lead to cognitive impairments [35]. Moreover, *SYNGAP1* mutant mice have been found to exhibit behavioral abnormalities that model symptoms of schizophrenia [36], which is noteworthy given the association between early cannabis use and increased risk for psychotic disorders [37]. Using Western blotting experiments, we found that the changes in SYNGAP1 were spatio-temporal and developmental in nature. Specifically, the upregulation of SYNGAP1 found in the synaptosomal fractions following 12 days of abstinence (i) was not present in the cytosolic fractions where SYNGAP1 is also known to localize [28,29], and (ii) was not present following only one day of drug abstinence. Moreover, (iii) we found that the changes in SYNGAP1 were present only when WIN was administered in adolescence and not in adulthood. SYNGAP1 plays a critical role in membrane trafficking of α-amino-3-hydroxy-5-methyl-4-isoxazolepropionic acid receptors (AMPARs), and neuronal overexpression of SYNGAP1 has been found to cause a significant reduction in the insertion of AMPARs into the plasma membrane [38]. This is in line with our previous studies, where we found decreases in synaptic AMPAR subunits following WIN exposure in rat adolescence that associated with increased behavioral sensitivity to cocaine [14].

When adolescent male rats were treated with a natural cannabinoid, i.e., THC, we found again an increase in SYNGAP1 levels in the PFC, demonstrating that the previously observed changes are not specific to synthetic cannabinoids. When assessing the effects of THC, we also examined additional brain regions, and found a pronounced upregulation of SYNGAP1 in the amygdala, but no significant changes in the dorsal striatum, hippocampus, or nucleus accumbens. Interestingly, besides the terms related to GABAergic processes, our GO analyses using the differentially expressed proteins also revealed a strong and highly significant enrichment for ‘amygdala development’ (GO:0021764; Appendix A). Moreover, a previous report that utilized a *SYNGAP1* haploinsufficiency mouse model, found evidence for hyperconnectivity in the amygdala of *SYNGAP1* mutant mice [39], further supporting a key role for SYNGAP1 in this brain region. Although our THC results suggest that the cannabinoid-induced effects on SYNGAP1 are brain region-specific, it cannot be ruled out that changes in SYNGAP1 still occur in cellular subpopulations of other brain regions that cannot be distinguished using crude extracts and Western blotting experiments.

In conclusion, to our knowledge, this is the first study to demonstrate that cannabinoid exposure results in an upregulation of SYNGAP1 that is spatio-temporal and developmental in nature. Specifically, cannabinoid-induced changes in SYNGAP1 were found to be present (i) only in the synaptic compartments (not in the cytosol), (ii) only following prolonged drug abstinence (not on abstinence day 1), and (iii) only following cannabinoid exposure in adolescence (not in adulthood). Moreover, our study provides proof-of-concept support for the use of subcellular fractionations as a method to enrich for cellular compartments and to increase the sensitivity for identifying molecular changes using downstream applications, such as quantitative proteomics. Nonetheless, there are limitations to our study that need to be acknowledged and addressed in future experiments, including: (i) the examination of SYNGAP1 changes following exposure to cannabinoids using female rats, (ii) the examination of SYNGAP1 changes following exposure to cannabinoids using contingent, e.g., self-administration, methods, and (iii) the identification of behavioral changes linked to the cannabinoid-induced upregulation of SYNGAP1. Moreover, additional studies are warranted to (iv) examine whether the changes in SYNGAP1 levels observed here in late adolescence also persist into adulthood. Since changes in SYNGAP1 have previously been linked to intellectual disabilities and other neurodevelopmental disorders in humans [21], the present study raises the possibility that SYNGAP1 may prove to be a key molecular mediator of the association between cannabis use in adolescence and adverse neuropsychiatric outcomes in adulthood [22].

## 4. Materials and Methods

### 4.1. Animals

Male Sprague Dawley rats obtained at PND 35 (adolescents) and PND 70 (adults) (ENVIGO, Italy) were fed standard rat chow and water ad libitum, and were housed in a climate-controlled animal room (5 per cage; 21 ± 2 °C; 60% humidity) under a 12 h light/dark cycle (lights on at 7 AM), as previously described [40]. The estimated timing of male rat adolescence was defined according to Schneider M. [24]. The mid-adolescent treatment period of PND 42–52, which coincides with male puberty, was chosen as a vulnerable developmental period for the consequences of cannabis exposure [23]. All animals were acclimated for one week before starting treatment with WIN 55,212-2 mesylate (WIN), Δ^9^-tetrahydrocannabinol (THC) or vehicle (i.e., at PND 42 for adolescents and at PND 77 for adults). WIN (Tocris, Bio-Techne Ltd., Abingdon, UK) was dissolved in 2% Tween-80, 2% ethanol, and 96% saline, and was injected intraperitoneally (IP) in a volume of 1 mL/kg of body weight. THC (RTI International, Research Triangle Park, NC, USA), 1 g per 5 mL in ethanol solution, was dissolved in vehicle containing 2% Tween-80, 2% ethanol, and 96% saline, and was injected IP in a volume of 1 mL/kg of body weight. Increasing doses of WIN (2 mg/kg, for 3 days; 4 mg/kg, for 4 days; 8 mg/kg, for 4 days), or THC (2.5 mg/kg, for 3 days; 5 mg/kg, for 4 days; 10 mg/kg, for 4 days) were given twice per day for 11 consecutive days. The protocols of cannabinoid administration, including dosages, were chosen according to previous literature reporting neurobehavioral changes with these drug regimens and to mimic heavy cannabis consumption in human adolescence [14,41,42,43,44]. Rats were sacrificed following forced drug abstinence of 1 or 12 days. Brain areas of interest were obtained by regional dissections, followed by immediate freezing in liquid nitrogen and storage at −80 °C until molecular processing. All animal procedures were carried out in compliance with the approved animal policies by the Ethical Committee for Animal Experiments at the University of Cagliari, and according to Italian (D.L. 26/2014) and European Council (63/2010) directives.

### 4.2. Synaptosomal and Cytosolic Extractions

To isolate synaptosomal and cytosolic fractions from brain regions of interest, the Syn-PER Synaptic Protein Extraction Reagent (Thermo Fisher Scientific, Waltham, MA, USA) was used. The isolated synaptosomes were lysed in the N-PER Neuronal Protein Extraction Reagent (Thermo Fisher Scientific) prior to any downstream applications, and all extraction reagents were supplemented with Pierce Protease and Phosphatase Inhibitor Mini Tablets (Thermo Fisher Scientific), as previously described [14]. We previously validated the efficacy of the synaptosomal/cytosolic extraction method by probing for proteins that are enriched in the corresponding fractions [14].

### 4.3. Synaptoproteomics

For synaptoproteomics, tandem mass tag (TMT)-based quantitative proteomics was used, as previously described [14,40]. In brief, Qubit Protein Quantification assays (Thermo Fisher Scientific) were used to determine the total protein yield from each synaptosomal sample. Proteins from 38 μg of serum filtrate were then purified by mini S-trap columns (Protifi, Farmingdale, NY, USA) and digested on column by trypsin. The Pierce Quantitative Fluorometric Peptide Assay (Thermo Fisher Scientific) was used to quantify peptide concentrations prior to TMT-labeling, and 25 μg of peptides were labeled with TMT 6-plex isobaric reagent and mixed for high pH reverse-phase peptide fractionation. The Orbitrap Fusion Tribrid Mass Spectrometer (Thermo Fisher Scientific) was used for MS/MS analysis with the MS3 data acquisition method. A total of five animals were run per group (N = 5 WIN-treated, N = 5 controls), including technical replications in triplicates. The Proteome Discoverer software (version 2.2; Thermo Fisher Scientific) was used to search the acquired MS/MS data against the Rat protein database downloaded from UniProt and to generate TMT ratios. Positive identification was set at a 5% peptide false discovery rate (FDR). Additionally, at least one unique peptide had to be identified per protein. Duplicated protein identifications from the database were removed. A total of 4951 proteins were quantified and included in the final data (see Appendix A).

### 4.4. Western Blotting

Western blotting experiments were performed according to standard procedures and as previously described [14]. In brief, protein concentrations of synaptosomal or cytosolic extracts were determined using the Pierce Detergent Compatible Bradford Assay Kit (Thermo Fisher Scientific) and equal amounts of sample were run on 4–20% Mini-PROTEAN TGX Precast Protein Gels (Biorad; Bio-Rad Laboratories Inc., Hercules, CA, USA) prior to transfer to Immobilon-FL PVDF membranes (EMD Millipore; EMD Millipore Corp., Billerica, MA, USA). Following antibody incubations and membrane washes, protein bands of interest were visualized on the Odyssey Classic fluorescent detection system (LI-COR, LI-COR Biotechnology, Lincoln, NE, USA). Images were acquired and quantified using the Image Studio Software (LI-COR). For blot re-probing, membranes were stripped using the Restore Fluorescent Western Blot Stripping Buffer (Thermo Fisher Scientific). The primary antibodies were: (i) SynGAP (D20C7) Rabbit mAb (#5539; Cell Signaling Technology, Inc., Danvers, MA, USA), (ii) Synaptophysin (D35E4) Rabbit mAb (#5461, Cell Signaling), and (iii) anti-beta Actin antibody [AC-15] (ab6276, Abcam; Abcam PLC, Cambridge, UK). SYNGAP1 data acquired from synaptosomal or cytosolic fractions were normalized to levels of Synaptophysin (SYP) or Actin, respectively, and then transformed to fold-change differences relative to control.

### 4.5. Statistics

For synaptoproteomics, the Qlucore Omics Explorer package (Qlucore, Lund, Sweden) was used to perform statistical analyses. TMT ratios (each channel/common reference) were calculated by the Proteome Discoverer software and normalized by the total peptide amount. The common reference was generated by pooling an equal amount of lysate from each sample. Log2 transformation and k-Nearest Neighbors (KNN) imputation were used for missing values. To identify differentially expressed proteins between the WIN-treated and the control groups, one-way Analysis of Variance (ANOVA) with Benjamini—Hochberg FDR correction was applied to the data, and statistical significance was set at q < 0.01. Differentially expressed proteins were subjected to gene ontology (GO) enrichment analysis with the PANTHER classification system using Fisher’s Exact test with FDR correction and q < 0.05 [45]. For Western blotting experiments, statistical analyses were performed using GraphPad Prism 8 (GraphPad Software, Inc., La Jolla, CA, USA), and statistical significance was set at *p* < 0.05. Two-group comparisons, between WIN-treated animals and controls, were performed using two-tailed unpaired Student’s *t*-test. Comparisons involving five different brain regions following exposure to THC were performed using two-way ANOVA followed by Sidak’s multiple comparisons test. Western blotting data are presented as mean values and error bars represent standard error of the mean (SEM). Putative outliers were identified using the ROUT test (q = 1%) and, if present, were excluded from the analyses. The number of biological and/or technical replicates used for each analysis is specified in the corresponding figure legends.

## Figures and Tables

**Figure 1 ijms-24-00698-f001:**
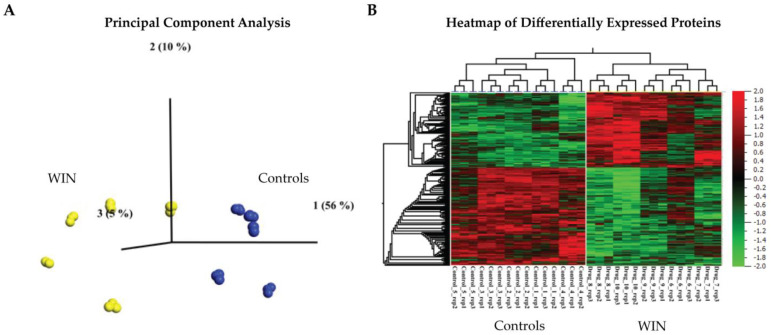
Synaptoproteomic analysis of the adolescent PFC following WIN-treatment. Male rats were exposed to a synthetic cannabinoid (WIN) or vehicle (i.e., controls) in mid-adolescence/puberty (PND 42–52) and proteomic analyses were applied to synaptosomal extracts from the prefrontal cortex (PFC) in late adolescence (PND 64) on the drug abstinence day (AD) 12 (N = 5 animals per condition and N = 3 technical replicates per animal in the proteomic analysis). (**A**) Supervised principal component analysis (PCA) following one-way Analysis of Variance (ANOVA) shows that 71% of the data are explained in PC1-3 for N = 487 differentially expressed proteins at q < 0.01. (**B**) Synaptoproteomics-derived heatmap of hierarchical cluster analysis of the N = 487 differentially expressed proteins in the PFC of WIN-treated animals versus controls (ANOVA, q < 0.01). The complete list of differentially expressed proteins is presented in Appendix A and the Gene Ontology (GO) results are presented in Appendix A.

**Figure 2 ijms-24-00698-f002:**
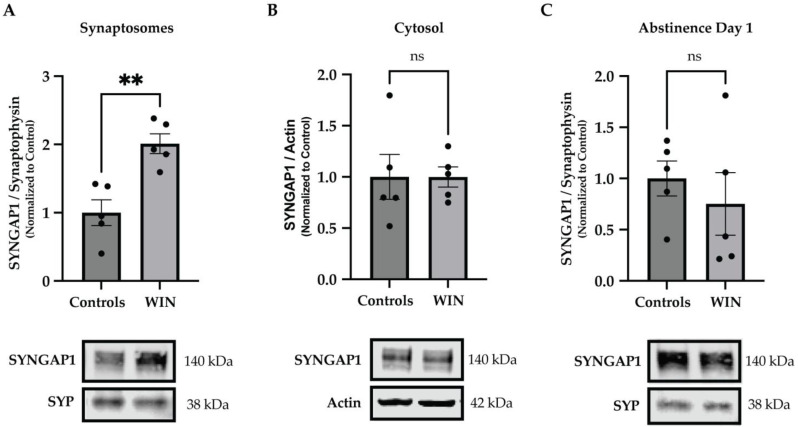
Exposure to WIN in adolescence results in upregulation of synaptic SYNGAP1 following drug abstinence. The ANOVA model applied to the synaptoproteomics data revealed that SYNGAP1 was among the most significant differentially expressed proteins in the PFC of WIN-treated animals compared to controls following 12 days of drug abstinence (ANOVA, q = 0.00000679; Appendix A). (**A**) Western blotting experiments using synaptosomal lysates of the PFC confirmed significant changes in levels of SYNGAP1 in the WIN-treated group compared to controls following 12 days of drug abstinence (n = 5 animals per group). (**B**) Western blotting experiments using cytosolic lysates of the PFC showed no changes in levels of SYNGAP1 in the WIN-treated group compared to controls following 12 days of drug abstinence (n = 5 animals per group), suggesting that the WIN-induced changes in SYNGAP1 are spatial in nature. (**C**) No changes were found in levels of SYNGAP1 when Western blotting experiments were performed using synaptosomal PFC extracts from animals sacrificed the day after the last WIN exposure, i.e., on abstinence day 1 (n = 5 animals per group), suggesting that the WIN-induced changes in SYNGAP1 are also temporal in nature. Graph data are presented as mean ± SEM. Representative Western blots are shown below the graphs, with the approximate molecular weights of observed band sizes indicated to the right. kDa, kilodaltons; ** *p* < 0.01, and ns = not significant.

**Figure 3 ijms-24-00698-f003:**
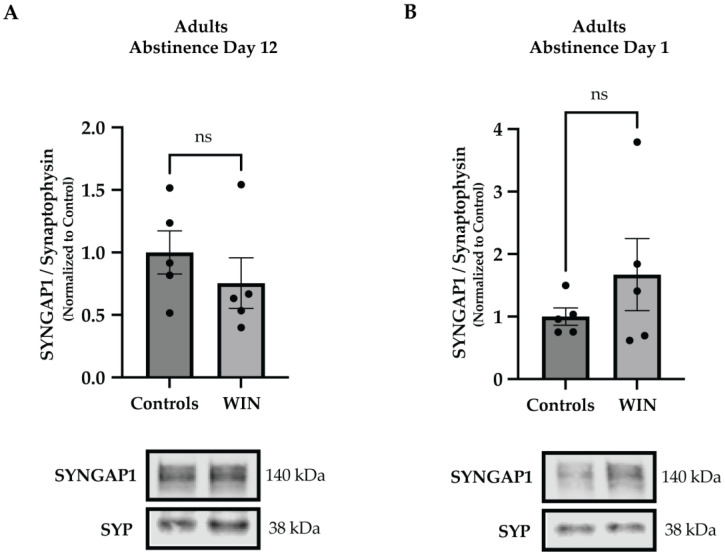
WIN administration in adulthood does not affect synaptic SYNGAP1 levels. Adult male rats were exposed to WIN (PND 77-87) and synaptosomal fractions from the PFC were isolated on drug abstinence days 12 and 1, mimicking the drug administration timeline and the analytic time points applied to adolescents. Western blotting analyses showed no changes in levels of SYNGAP1 on (**A**) abstinence day 12 or on (**B**) abstinence day 1 (n = 5 animals per group), suggesting that the WIN-induced changes in SYNGAP1 are specific to adolescence. Graph data are presented as mean ± SEM. Representative Western blots are shown below the graphs, with the approximate molecular weights of observed band sizes indicated to the right. kDa, kilodaltons; and ns = not significant.

**Figure 4 ijms-24-00698-f004:**
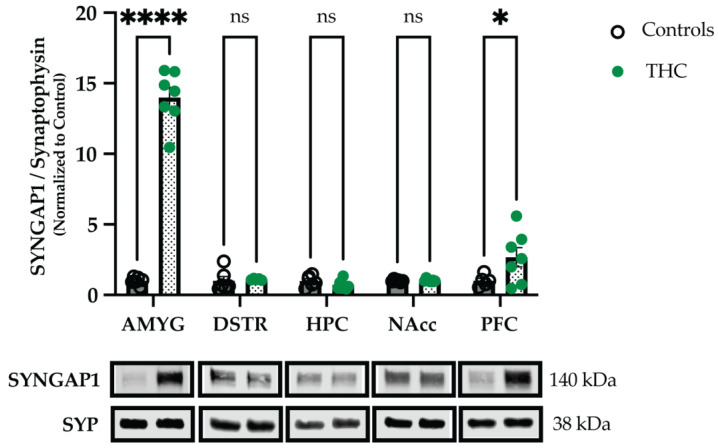
Exposure to THC in adolescence affects levels of SYNGAP1 in the amygdala and PFC. Male rats were exposed to Δ^9^-tetrahydrocannabinol (THC) or vehicle (i.e., controls) in mid-adolescence/puberty (PND 42–52) and synaptosomal fractions were extracted from five different brain regions in late adolescence (PND 64) on drug abstinence day 12. Western blotting analyses showed significant upregulation in synaptic levels of SYNGAP1 in the amygdala and prefrontal cortex of THC-treated animals (N = 6–7 animals per group; N = 1 outlier in DSTR and N = 1 outlier in PFC). Graph data are presented as mean ± SEM. Representative Western blots are shown below the graphs, with the approximate molecular weights of observed band sizes indicated to the right. AMYG, amygdala; DSTR, dorsal striatum; HPC, hippocampus; kDa, kilodaltons; NAcc, nucleus accumbens; PFC, prefrontal cortex. * *p* < 0.05, **** *p* < 0.0001, and ns = not significant.

## Data Availability

All data have been uploaded as Appendix A.

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
