# Peer review of "Synaptoproteomic Analysis of the Prefrontal Cortex Reveals Spatio-Temporal Changes in SYNGAP1 Following Cannabinoid Exposure in Rat Adolescence"

_ijms, 2022, doi:10.3390/ijms24010698_

Round 1
Reviewer 1 Report
Eliminate the word Figure from the figures, these are only placed in the figure caption. The font in the figures is different from the text.
Correct the format for figure captions (it is not homogeneous)
Line 118. Use the same format to name the supplementary data (Supplementary Dataset 2 or Supplementary Dataset S2).
Line 160, 204, 205, 206. Check font
Line 232. Check spaces in citations
Line 291. Check citation format
Line 297, 298, 300. Change ml by mL
Line 330. Place equipment's origin
Line 397. The font of the references is different
Reviewer 2 Report
In the manuscript ‘Synaptoproteomic analysis of the prefrontal cortex reveals Spatio-temporal changes in SYNGAP1 following cannabinoid exposure in rat adolescence’, the authors have claimed it as the first study to demonstrate the relationship between cannabinoid exposure to changes in SYNGAP1. It is interesting to know that cannabinoid exposure led to the upregulation of SYNGAP 1, which was spatiotemporal and developmental in nature. The study demonstrates a further need for work in this area. The manuscript is well written and has conveyed the result. The article is recommended for publication. However, there are some suggestions for the authors- The study would have been more validated and accepted if the authors had performed behaviour assays to substantiate the neuro-behavioural changes associated with cannabinoid exposure in different age groups. I recommend that the authors upload the proteomics data to databases such as PRIDE etc.
